# BotaCLIP: Contrastive Learning for Botany-Aware Representation of Earth Observation Data

## Abstract

Foundation models have demonstrated a remarkable ability to learn rich, transferable representations across diverse modalities such as images, text, and audio. In modern machine learning pipelines, these representations often replace raw data as the primary input for downstream tasks. In this paper, we address the challenge of adapting a pre-trained foundation model to inject domain-specific knowledge, without retraining from scratch or incurring significant computational costs. To this end, we introduce BotaCLIP, a lightweight multimodal contrastive framework that adapts a pre-trained Earth Observation foundation model (DOFA) by aligning high-resolution aerial imagery with botanical relevés. Unlike generic embeddings, BotaCLIP internalizes ecological structure through contrastive learning with a regularization strategy that mitigates catastrophic forgetting. Once trained, the resulting embeddings serve as transferable representations for downstream predictors. Motivated by real-world applications in biodiversity modeling, we evaluated BotaCLIP representations in three ecological tasks: plant presence prediction, butterfly occurrence modeling, and soil trophic group abundance estimation. The results showed consistent improvements over those derived from DOFA and supervised baselines. More broadly, this work illustrates how domain-aware adaptation of foundation models can inject expert knowledge into data-scarce settings, enabling frugal representation learning.

## 1 Introduction

Plants form the foundation of terrestrial ecosystems, driving primary productivity and supporting the diversity of nearly all other life forms (Cavender-Bares et al., 2020). Vegetation integrates ecological characteristics such as soil, microclimate, and species assemblages (Chauvier et al., 2021), and serves as a key proxy for understanding ecosystem functioning and biodiversity patterns across scales (Walker & Wardle, 2014; Ibarra-Manriquez et al., 2022). Beyond ecology, vegetation dynamics are central to climate change mitigation and conservation planning. However, ecological data such as vegetation surveys, also known as relevés (tabular records of species occurrence and coverage) are rich but spatially sparse, while Earth Observation (EO) imagery provides global coverage yet is often too generic to capture fine-scale biological signals. Recent EO foundation models (Xiong et al., 2024; Szwarcman et al., 2024; Wang et al., 2025) have demonstrated strong transfer across tasks such as land-cover classification, canopy height estimation, and temporal monitoring, highlighting the potential of generic embeddings as standard inputs for downstream predictors. Yet, despite these advances, such representations remain insufficiently specialized for ecological applications, as they rarely align with species composition or community structure, limiting their usefulness for biodiversity modeling and climate-relevant forecasting.

Contrastive learning (CL) has emerged as a powerful tool for bridging heterogeneous modalities. CLIP (Radford et al., 2021) pioneered large-scale image–text pretraining, inspiring extensions to tabular–image settings such as TIP (Du et al., 2024) and to satellite image-metadata (Bourcier et al., 2024). These works highlight that contrastive objectives allow to embed auxiliary modalities (metadata and tabular data) into visual representations, enriching them with semantic context. Compared to supervised multimodal fusion, which requires task-specific labels and often struggles with incomplete or imbalanced data, contrastive approaches leverage weak supervision from paired samples and

yield more transferable embeddings (a property confirmed in our experiments). Methodologically, contrastive learning can also be seen as a non-linear extension of Canonical Correspondence Analysis (CCA) (ter Braak, 1986), long used in ecology to relate species composition to environmental gradients. Yet, despite vegetation plots being one of the richest ecological data sources, no contrastive framework to date has aligned EO imagery with large-scale relevé data.

In this paper, we introduce BotaCLIP, a lightweight, botany-aware multimodal framework that adapts DOFA EO foundation model embeddings by aligning high-resolution aerial images with vegetation relevés via contrastive learning. To preserve the generalization ability of EO encoders, we propose a regularization strategy that mitigates catastrophic forgetting by maintaining the local similarity structure from the foundation embeddings. This lightweight design enables scalable integration of ecological knowledge without expensive end-to-end training.

Our work provides both domain and machine learning contributions:

- We demonstrate that image embeddings obtained through contrastive alignment outperform both original foundation model embeddings and those derived from supervised baselines, underscoring their value for ecological prediction.
- We show that fine-tuning large encoders may be avoidable, as lightweight embedding post-processing already delivers performance across diverse downstream tasks (plants, insects, and soil monitoring).
- We highlight the role of regularization in preserving general representation quality while enriching embeddings with domain-specific semantics.
- We deliver an inexpensive pipeline for adapting foundation models, consistent with modern machine learning best practices of specialization on top of efficient pretrained backbones.

BotaCLIP illustrates how simple domain-aware alignment allows to bootstrap downstream performance. We believe our framework will benefit to all practitioners that need specialized representations but want a lightweight framework for fast experimentation, which is relevant well beyond biodiversity modeling.

## 2 THE BOTACLIP FRAMEWORK

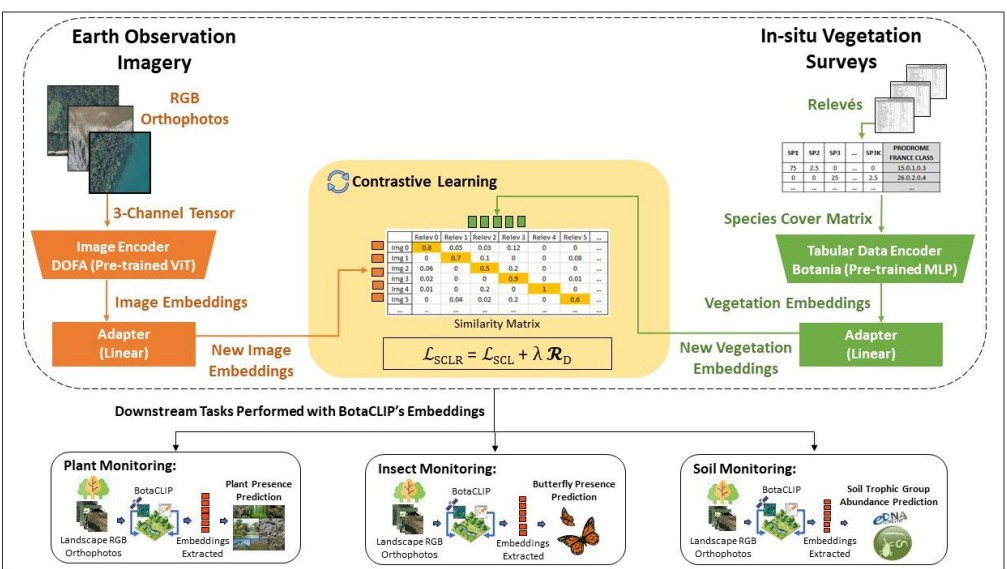

Figure 1: Overview of the BotaCLIP framework. RGB orthophotos are encoded with the pretrained ViT model DOFA and vegetation relevés with the pre-trained MLP model Botania. The two modalities are aligned with a contrastive objective regularized by the similarity structure of DOFA embeddings. After training, BotaCLIP embeddings are extracted from the image adapter using new orthophotos and serve as inputs for downstream tasks in plant, insect, and soil monitoring.

BotaCLIP is a multimodal pipeline that aligns EO imagery with in-situ vegetation surveys (Figure 1). Aerial RGB orthophotos are encoded with the EO foundation model DOFA, yielding generic visual embeddings, while relevés are transformed into species–cover matrices and encoded with a pre-trained MLP that we call *Botania*. Both streams pass through linear adapters and are projected into a shared latent space, where paired samples are aligned with a sigmoid contrastive loss ($\mathcal{SCL}$) regularized by DOFA similarities ($\lambda \mathcal{R}_{\mathcal{D}}$). This prevents catastrophic forgetting and enriches EO-derived embeddings with botanical semantics without sacrificing their general representational capacity.

The resulting space produces complementary image- and tabular-based representations. In practice, we focus on image embeddings for downstream evaluation in plant, insect, and soil monitoring. This choice reflects a pragmatic consideration: vegetation surveys provide rich ecological information but are costly and time-intensive to collect, while high-resolution aerial imagery is widely available and scalable. Image-based embeddings thus offer the most realistic entry point for biodiversity applications at large spatial scales.

## 2.1 Data Modalities and Preprocessing

The BotaCLIP framework integrates two data types:

**Earth Observation Imagery**. We used high-resolution aerial orthophotographs from the BD ORTHO® dataset (national de l'information géographique et forestière , IGN) (IGN), geometrically rectified and updated every 3–4 years at 20cm resolution. For each vegetation plot (30m × 30m), we extracted a 100m×100m orthophoto, yielding 28,418 RGB images. These were processed with DOFA (Xiong et al., 2024), a ViT-based EO foundation model pretrained on multispectral, hyperspectral, and SAR data. Here we used only RGB inputs, resized to $224 \times 224$, normalized with dataset-specific statistics, and extracted 768-dimensional embeddings from the penultimate layer.

**In-situ Vegetation Surveys**. The second modality comprises 28,418 relevés from the Conservatoire Botanique National Alpin (CBNA), reporting the abundance of 3,587 plant species as tabular data using the Braun-Blanquet cover-abundance scale. The Braun-Blanquet classes were converted to mean percentage values, harmonizing field estimates into continuous inputs. Each relevé was assigned to one of 232 vegetation classes in the *Prodrome des Végétations de France* (Bardat et al., 2001), forming a species-by-plot cover matrix (28,418 × 3,588) with an associated categorical label.

To derive tabular features, we pretrained *Botania*, a lightweight MLP for phytosociological classification. It takes the 3,587-dimensional species–cover vector and predicts vegetation class:

$$3587 \xrightarrow{\text{Linear}} 1536 \xrightarrow{\text{GELU}} \xrightarrow{\text{Dropout(0.4)}} \xrightarrow{\text{Linear}} 768 \xrightarrow{\text{GELU}} \xrightarrow{\text{Dropout(0.4)}} \xrightarrow{\text{Linear}} 232,$$

with a normalized 768-dimensional representation extracted from the penultimate layer. Botania was trained with 300 epochs with Adam (lr = 0.3, patience = 20), reaching 66% top-1 and 86% top-3 accuracy. These embeddings were used for contrastive alignment in BotaCLIP.

Each relevé is georeferenced, enabling pairing with its orthophoto. This spatial linkage provides aligned image–tabular samples for training.

## 2.2 Architecture and Contrastive Objective

**Images.** As stated above, we do not directly work on raw images, but on their DOFA embeddings, which we denote $\text{Img}_i \in \mathbb{R}^{768}$. These embeddings are processed by a lightweight adapter $A^{\text{img}}$ with learnable parameters $\theta_{\text{img}}$. In our configuration, this adapter is implemented as a Linear layer mapping $768 \to 768$. To initialize this adapter, we set its weights to the identity matrix and add a small Gaussian perturbation of variance $10^{-4}$, while the bias is set to zero. This ensures that the adapter starts close to an identity mapping, preserving DOFA embeddings at initialization, while introducing enough noise to break symmetry and allowing the adapter to learn domain-specific transformations.

**Vegetation.** On the vegetation side, species–cover vectors are processed by Botania, which outputs 768-dimensional embeddings $\text{Tab}_i \in \mathbb{R}^{768}$ from its penultimate hidden layer. As for images, we apply a lightweight adapter $A^{\text{tab}}$ with learnable parameters $\theta_{\text{tab}}$, implemented as a Linear layer mapping $768 \to 768$. Unlike the image branch, no identity initialization is required; the adapter is initialized with default PyTorch settings.

The final projected embeddings are denoted $z_i^{\mathrm{img}} = A^{\mathrm{img}}(\mathrm{Img}_i) \in \mathbb{R}^{768}$ for the image branch and $z_i^{\mathrm{tab}} = A^{\mathrm{tab}}(\mathrm{Botania}(\mathrm{Tab}_i)) \in \mathbb{R}^{768}$ for the tabular branch. Both outputs are $\ell_2$-normalized to lie on the unit hypersphere and projected into the shared embedding space for alignment via CL.

**Sigmoid contrastive loss.** At its core, BotaCLIP relies on the sigmoid contrastive loss (Zhai et al., 2023) to align paired image–relevé samples while contrasting mismatches. Given a batch of $N$ pairs, we use the projected embeddings defined above $z_i^{\mathrm{img}}$ and $z_i^{\mathrm{tab}}$. For two vectors $z, z' \in \mathbb{R}^{768}$, let $z \cdot z'$ denote their scalar product. Pairwise logits are then computed as:

$$\ell_{ij}(\theta) = (\, z_i^{\mathrm{img}} \cdot z_j^{\mathrm{tab}} \,) \exp(\tau) + b, \tag{1}$$

where $\tau$ is a learnable temperature, $b$ a learnable bias, and $\theta = (\theta_{\mathrm{img}}, \theta_{\mathrm{tab}}, \tau, b)$ collects all learnable parameters. We construct labels $\omega_{ij} = +1$ for positive pairs ($i = j$) and $\omega_{ij} = -1$ otherwise. Then, being $\sigma(\cdot)$ the logistic sigmoid, the sigmoid contrastive loss is:

$$\mathcal{L}_{\mathrm{SCL}}(\theta) = -\frac{1}{N^2} \sum_{i=1}^{N} \sum_{j=1}^{N} \log \sigma\big(\omega_{ij}\, \ell_{ij}(\theta)\big), \tag{2}$$

**Regularization.** Since the image embeddings $\mathrm{Img}_i$ are extracted from a pretrained encoder (DOFA), they already contain meaningful semantic structure. Our goal is to enrich them with vegetation information without discarding this prior knowledge. Relying solely on the contrastive loss $\mathcal{L}_{\mathrm{SCL}}$ can lead to *catastrophic forgetting* (McCloskey & Cohen, 1989). Mathematically, the optimization drives $z_i^{\mathrm{img}}$ to match $z_i^{\mathrm{tab}}$, reshaping the image space around dimensions that distinguish relevés while collapsing others that carry no gradient signal. Ecologically, this means that cues captured by DOFA but not strongly linked to vegetation composition (e.g., soil, relief, or anthropogenic patterns) risk being discarded, reducing the transferability of the embeddings to broader EO tasks.

To mitigate this, we introduce a regularization term that encourages the projected embeddings $z_i^{\mathrm{img}}$ to preserve the local similarity structure of the original DOFA embeddings $\mathrm{Img}_i$. Rather than enforcing $z_i^{\mathrm{img}} \approx \mathrm{Img}_i$ directly, we constrain pairs that were close in DOFA space to remain close after projection. Formally, we define:

$$\mathcal{R}(\theta) = \frac{1}{N^2} \sum_{i=1}^{N} \sum_{j=1}^{N} W_{ij} \left( \mathrm{Img}_i \cdot \mathrm{Img}_j - z_i^{\mathrm{img}} \cdot z_j^{\mathrm{img}} \right)^2, \tag{3}$$

where $W_{ij} = \left(\frac{1 + \mathrm{Img}_i \cdot \mathrm{Img}_j}{2}\right)^2$ assigns higher weight to pairs that are similar in DOFA space. This strategy specializes the embeddings while preserving neighborhood relations, akin in spirit to manifold-preserving methods such as UMAP (McInnes et al., 2018). The regularization is computationally lightweight, requiring only dot products between already computed embeddings. The final training objective combines contrastive alignment with this regularization, where $\lambda > 0$ controls its strength:

$$\mathcal{L}_{\mathrm{SCLR}}(\theta) = \mathcal{L}_{\mathrm{SCL}}(\theta) + \lambda\, \mathcal{R}_{\mathcal{D}}(\theta), \tag{4}$$

## 2.3 TRAINING STRATEGY

BotaCLIP is trained with spatial cross-validation to avoid leakage due to spatial autocorrelation (Roberts et al., 2017). The study region is partitioned into 5km $\times$ 5km grid cells (ETRS89/LAEA, EPSG:3035), and each relevé is assigned to its corresponding cell. Folds are defined at the cell level, with an additional one-cell buffer around each validation fold to ensure that training samples are at least 5 km away from validation samples. For efficiency, we used a single fold ($k = 1$), which both preserves spatial separation and reflects the practical need for downstream tasks to rely on a specific checkpoint rather than averaged models.

To improve robustness, we applied standard image augmentations to the training set (random flips, 90°/270° rotations, color jitter, Gaussian blur, and random resized cropping), while keeping validation images unchanged. Multiple augmented views of each orthophoto were paired with the same relevé, enlarging the training set and increasing invariance to viewpoint, illumination, and texture variations. Although such invariances are partly encoded in the foundation model, we found augmentations still marginally improved embedding quality.

Optimization used AdamW with learning rate $10^{-3}$, weight decay $10^{-3}$, batch size 256 and the regularization coefficient fixed to $\lambda = 1$, training for up to 1000 epochs with early stopping (patience $= 10$). The DOFA backbone remained frozen, so only the lightweight adapters and the tabular encoder were updated. Botania, in contrast, was initialized from its pre-trained checkpoint but kept trainable, allowing its representations to adapt jointly with the contrastive objective. With projection dimension 768, this setup trains $\sim$8.1M parameters versus $\sim$111M for DOFA, avoiding recomputation of patch-level embeddings and making training inexpensive in both compute and memory. Our aim is not to release another foundation model, but to provide a practical methodology for adapting existing EO encoders with ecological knowledge, making BotaCLIP lightweight and accessible. Additional ablation studies on architectural and loss variants are reported in Section 4.

## 3 EXPERIMENTAL SETUP

### 3.1 BASELINES

We compare BotaCLIP against two alternatives: raw DOFA embeddings and a supervised pre-training baseline (BotaSP). DOFA embeddings serve as the simplest reference, while BotaSP trains a linear projection and MLP classifier on plant presence/absence labels using DOFA embeddings as input (proj. 768, hidden 1536, GELU, Dropout 0.4). The model is optimized with AdamW (lr=0.001, wd=0.001, batch=256) for 200 epochs with early stopping, using a loss $\mathcal{L} = \mathcal{L}_{\text{CE}} + \lambda \, \mathbb{E}[W \cdot (S_{\text{new}} - S_{\text{orig}})^2]$, where $\mathcal{L}_{\text{CE}}$ is cross-entropy, $S_{\text{orig}} = zz^\top$ and $S_{\text{new}} = z'z'^\top$ are pairwise similarities before and after projection, $W = ((1 + S_{\text{orig}})/2)^2$ are similarity-based weights, and $\lambda = 100$. After training, the classification head is discarded and penultimate features are used for downstream tasks.

### 3.2 DOWNSTREAM TASKS

All baselines and BotaCLIP embeddings were evaluated on three applications: plant, insect, and soil biodiversity monitoring. In all cases, species or trophic-group labels were georeferenced and paired with BD ORTHO® aerial photographs (20cm resolution, cropped to $100 \times 100$ m), from which image embeddings were extracted.

Downstream models are Random Forests from Scikit-learn (Pedregosa et al., 2011) with default hyper-parameters. For plants and insects, experiments were repeated over 10 seeds with Stratified K-Fold cross-validation ($K = 1$ for plants, $K = 5$ for butterflies); for soil, we used 5-fold CV. Results are averaged across seeds and folds. We chose this simple pipeline to match common ecological practice, which relies on libraries such as BioMod2 (Guéguen et al., 2025). This also ensures that performance differences reflect embedding quality rather than downstream model complexity.

**Metrics.** For standard evaluation we report F1, Sensitivity (Sensi.), Mean Absolute Error (MAE), and Spearman's $\rho$. For species distribution tasks, we also include two ecological metrics: the *Boyce Index* (BI), which measures how well predicted presences match observed spatial distributions beyond random expectation (Broennimann et al., 2025), and the *True Skill Statistic* (TSS), which combines sensitivity and specificity and is widely used to assess presence–absence models (Allouche et al., 2006).

**Plant Monitoring: Plant Presence Prediction**.
*Dataset*: We used the same set of 28,418 relevés from the French Alps (3,587 species) employed to train BotaCLIP. This task is not a retraining of the model, but an explicit test of transfer: we evaluate whether image embeddings alone retain the botanical information aligned from relevés during contrastive learning. Species–cover values were binarized into presence (value $> 0$) or absence ($= 0$), yielding true absence information unlike pseudo-absence strategies (when we don't know if the species was actually missing or just not observed). To ensure sufficient support, we retained only species with at least 1,000 presences. Following the spatial split defined for BotaCLIP, we used fold $k = 1$ to keep training and validation spatially disjoint. To balance classes, absences were downsampled to match presences in both sets.
*Target*: Predict binary presence/absence labels for each plant species.
*Metrics*: TSS, F1, and Sensitivity.

**Insect Monitoring: Butterfly Presence Prediction**.

*Dataset*: Butterfly occurrence records were compiled from GBIF, restricted to human observations (2000–2022) with spatial precision $\leq 1$ km, and cleaned with the CoordinateCleaner R package (Zizka et al., 2019). We retained only records within the French Alps, discarding those below 250m elevation (urban/industrial areas) and species with fewer than 100 or more than 1,000 presences, keeping 134 species in total. The former lack statistical power, while the latter are highly generalist and ubiquitous, making their presence hard to predict from local imagery. Restricting to this intermediate range yields species with sufficient data and stronger ecological signal. Presence/absence datasets were built using pseudo-absences: occurrences marked as presences, and all other coordinates as candidate absences, downsampled to match presences for class balance. We applied a spatial 5-fold split with 5 km cells and a 1-cell buffer to avoid leakage.
*Target*: Predict binary presence/absence labels for each butterfly species.
*Metrics*: TSS, BI, F1, and Sensitivity.

**Soil Monitoring: Soil Trophic Group Abundance Prediction**.

*Dataset*: We used soil eDNA data from the French Alps long-term observatory OR-CHAMP (Thuiller, 2024), as detailed in (Calderón-Sanou et al., 2022). Between 2016 and 2020, 953 soil samples were collected across 26 elevational gradients and processed with multi-marker DNA metabarcoding, yielding relative abundances for 51 trophic groups spanning biological categories (Bacteria, Fungi, Protist, Oligochaete, Insect, Collembola, Metazoa). Abundances were normalized within samples (relative proportions) and across samples (min–max scaling), and samples were stratified by elevation quantiles before cross-validation to preserve altitudinal distributions.
*Target*: Predict continuous abundances per trophic group.
*Metrics*: MAE and Spearman's $\rho$.

## 3.3 ABLATION STUDIES

To systematically explore the design space of BotaCLIP, we defined a compact naming scheme in which each variant is identified by concatenating three components:

**Architecture:** B = Botania encoder, M = MLP encoder, A = Attention-based encoder.
**Augmentation:** WiAu = trained with image augmentation, WoAu = trained without augmentation.
**Objective:** Scl = sigmoid contrastive loss, SclR = our regularized sigmoid contrastive loss.

For example, BWiAuSclR denotes the Botania encoder with augmentation and the regularized loss, while MWoAuScl refers to an MLP adapter without augmentation under the plain loss.

We investigated these axes for the following reasons. First, we included a simple MLP encoder as a baseline (MWiAuScl, MWoAuScl), since MLPs remain a competitive choice for small tabular models. Second, we tested a Multihead Attention block on the tabular branch (AWiAuScl, AWoAuScl), motivated by the potential of attention to capture interactions across heterogeneous features in ecological data. Third, we considered Botania (BWiAuSclR, BWoAuSclR), a streamlined tabular encoder that leverages ecological priors to better capture vegetation structure and landscape composition. Finally, we contrasted the role of data augmentation and of our proposed loss regularization in shaping the learned representations.

Detailed experimental setups are in Appendix A.1, while summarized results are in Section 4.

## 4 RESULTS

### 4.1 DOWNSTREAM PERFORMANCE AND ABLATIONS

Table 1 reports the performance of all BotaCLIP variants together with the DOFA and BotaSP baselines. Mean and standard deviation are computed over seeds and folds, allowing us to assess both accuracy and stability of each configuration. Overall, models based on the Botania encoder and trained with our regularized contrastive loss outperform both DOFA and BotaSP, though the difference between the two Botania variants (with vs. without augmentation) is not immediately evident from mean values alone.

To resolve the ambiguity between Botania variants, Table 1 reports not only mean $\pm$ std but also three additional rows per task: Best model, Friedman $p$-val, and $\Delta$ vs. DOFA. The latter expresses

Table 1: Ablation study across BotaCLIP variants. Metrics are reported as mean $\pm$ std (over seeds and folds). DOFA and BotaSP are included as baselines. Additional rows report statistical analysis (Friedman and Wilcoxon-Holm).

| Dataset | Metric | DOFA | BotaSP | BWiAuSclR | BWoAuSclR | MWiAuScl | MWoAuScl | AWiAuScl | AWoAuScl |
|---|---|---|---|---|---|---|---|---|---|
| Plant | TSS | $0.42 \pm 0.00$ | $0.47 \pm 0.00$ | $\mathbf{0.49 \pm 0.00}$ | $\mathbf{0.49 \pm 0.00}$ | $0.42 \pm 0.00$ | $0.44 \pm 0.00$ | $0.41 \pm 0.00$ | $0.41 \pm 0.00$ |
| | F1 | $0.24 \pm 0.00$ | $0.26 \pm 0.00$ | $\mathbf{0.27 \pm 0.00}$ | $\mathbf{0.27 \pm 0.00}$ | $0.23 \pm 0.00$ | $0.24 \pm 0.00$ | $0.23 \pm 0.00$ | $0.24 \pm 0.00$ |
| | Sens. | $0.71 \pm 0.00$ | $0.73 \pm 0.00$ | $\mathbf{0.74 \pm 0.00}$ | $\mathbf{0.74 \pm 0.00}$ | $0.73 \pm 0.00$ | $0.73 \pm 0.00$ | $0.72 \pm 0.00$ | $0.69 \pm 0.00$ |
| Best model | | BWiAuSclR (Wilcoxon-Holm, $p < 10^{-19}$) | | | | | | | |
| Friedman $p$-val | | $3.9 \times 10^{-105}$ | | | | | | | |
| $\Delta$ vs DOFA | | $+14.9\%$ (median TSS) | | | | | | | |
| Butterfly | TSS | $0.29 \pm 0.01$ | $0.31 \pm 0.01$ | $\mathbf{0.33 \pm 0.01}$ | $\mathbf{0.33 \pm 0.01}$ | $0.29 \pm 0.01$ | $0.30 \pm 0.01$ | $0.27 \pm 0.01$ | $0.27 \pm 0.01$ |
| | BI | $0.66 \pm 0.03$ | $0.68 \pm 0.03$ | $0.70 \pm 0.02$ | $\mathbf{0.71 \pm 0.03}$ | $0.60 \pm 0.03$ | $0.62 \pm 0.03$ | $0.56 \pm 0.03$ | $0.63 \pm 0.03$ |
| | F1 | $0.68 \pm 0.01$ | $0.69 \pm 0.01$ | $\mathbf{0.70 \pm 0.01}$ | $\mathbf{0.70 \pm 0.01}$ | $0.69 \pm 0.01$ | $0.69 \pm 0.01$ | $0.69 \pm 0.01$ | $0.68 \pm 0.01$ |
| | Sens. | $0.77 \pm 0.01$ | $0.78 \pm 0.01$ | $0.79 \pm 0.01$ | $0.79 \pm 0.01$ | $\mathbf{0.80 \pm 0.01}$ | $\mathbf{0.80 \pm 0.01}$ | $\mathbf{0.80 \pm 0.01}$ | $0.76 \pm 0.01$ |
| Best model | | BWiAuSclR (Wilcoxon-Holm, $p < 10^{-22}$) | | | | | | | |
| Friedman $p$-val | | $8.8 \times 10^{-107}$ | | | | | | | |
| $\Delta$ vs DOFA | | $+10.4\%$ (median BI) | | | | | | | |
| Soil | MAE | $0.12 \pm 0.05$ | $0.12 \pm 0.05$ | $0.12 \pm 0.05$ | $0.12 \pm 0.05$ | $0.12 \pm 0.05$ | $0.12 \pm 0.05$ | $0.12 \pm 0.05$ | $0.12 \pm 0.05$ |
| | Spear. $\rho$ | $0.40 \pm 0.15$ | $0.40 \pm 0.14$ | $0.41 \pm 0.15$ | $\mathbf{0.41 \pm 0.14}$ | $0.41 \pm 0.15$ | $\mathbf{0.41 \pm 0.14}$ | $0.41 \pm 0.14$ | $0.40 \pm 0.15$ |
| Best model | | BWiAuSclR (Wilcoxon-Holm, $p = 4.6 \times 10^{-4}$ vs DOFA) | | | | | | | |
| Friedman $p$-val | | $9.3 \times 10^{-5}$ | | | | | | | |
| $\Delta$ vs DOFA | | $+1.8\%$ (median $\rho$) | | | | | | | |

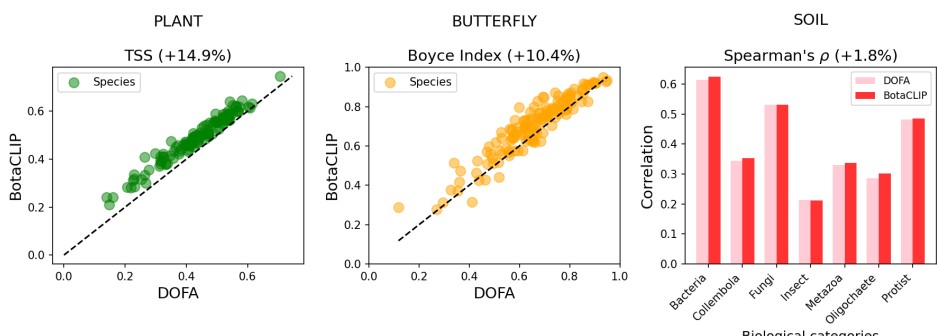

Figure 2: Performance of DOFA vs. BotaCLIP on plant (TSS), butterfly (BI), and soil (Spearman's $\rho$) tasks. Scatter plots (left, middle) show per-species scores with the identity line as reference. The bar plot (right) shows mean correlations by trophic groups aggregated by biological categories. $\% \uparrow$ denotes average relative gain of BotaCLIP over DOFA.

the relative improvement of the best configuration over DOFA, measured on the representative metric of each task. For plants, we focus on TSS, as true presence–absence labels are available and this statistic provides a balanced evaluation of commission and omission errors under class imbalance. For butterflies, we report BI, as evaluation relies on presence–only data with pseudo-absences, making habitat suitability ranking the appropriate criterion. For soil trophic groups, we use Spearman's $\rho$, as the goal is to recover the relative abundance structure across functional categories rather than exact absolute values. This design follows common practice in ecological evaluation, where statistical tests are carried out at the per-species (or per-group) level.

We used a Friedman test to assess whether global differences exist across models, followed by paired Wilcoxon signed-rank tests with Holm–Bonferroni correction against DOFA, as it represents the unaligned embeddings whose improvement we seek to quantify. The analysis identifies BWiAuSclR (Botania with augmentation and regularized loss) as the best configuration, yielding systematic gains over DOFA of $+14.9\%$ (plants, TSS), $+10.4\%$ (butterflies, BI), and $+1.8\%$ (soil, $\rho$). We refer to configuration BWiAuSclR simply as BotaCLIP in the remainder of the paper.

Figure 2 provides a species-level view of the gains summarized in Table 1. For plants, nearly all points lie above the diagonal, indicating that BotaCLIP improves TSS consistently across species, not just on average. For butterflies, the upward shift in the cloud of points confirms higher BI values for most species, reflecting improved ability to rank habitat suitability from presence-only

data. For soil trophic groups, the bar plots reveal smaller but systematic increases in Spearman's $\rho$ across functional categories. These visualizations corroborate the median improvements reported in Table 1, showing that the observed gains are broadly distributed across taxa rather than driven by a few outliers.

## 4.2 Embedding Space Analysis

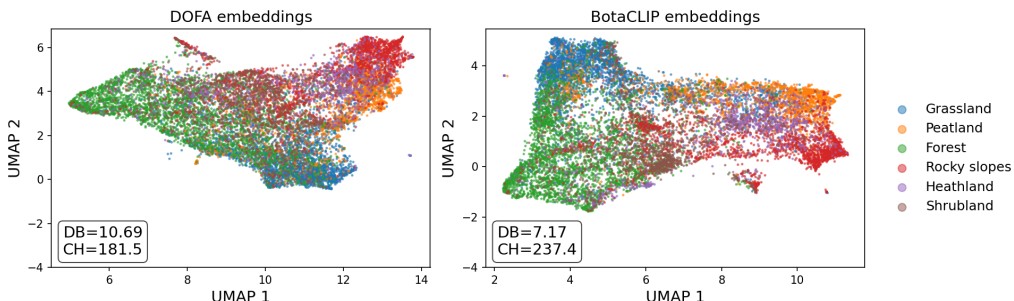

Figure 3: UMAP 2D visualization of DOFA (left) and BotaCLIP (right) embeddings, colored by six broad landscape categories.

We analyze BotaCLIP embedding space to examine whether the quantitative gains observed in downstream tasks also manifest in the structure of the learned representations. Embeddings were projected onto two dimensions using UMAP (McInnes et al., 2018). For interpretability, the 232 vegetation classes of the Prodrome were grouped by expert inspection into six broad landscape categories (Forests, Grasslands, Heathlands, Peatlands, Rocky slopes, Shrublands). These categories were not used during training or evaluation, but only as an external reference for visualization.

Figure 3 contrasts DOFA and BotaCLIP embeddings. DOFA already separates broad clusters despite never being exposed to these categories. BotaCLIP further sharpens the structure, with clearer boundaries for broad landscape categories. We further quantify cluster quality using the Davies–Bouldin (DB) and Calinski–Harabasz (CH) indices. BotaCLIP achieves a lower DB index (7.17 vs. 10.69) and a higher CH index (237.4 vs. 181.5).

## 5 Discussion

The ablation study revealed that architecture, augmentation, and loss design each shape the quality of BotaCLIP embeddings, but the dominant factor is the Botania encoder trained with our regularized loss. Among BotaCLIP variants, BWiAuSclR consistently emerged as the best model, with significant improvements over DOFA. To contextualize these results, we also introduced BotaSP, a supervised pretraining baseline in which DOFA embeddings were trained directly on plant presence/absence. This setup formalizes the natural alternative of supervised pretraining, predicting relevés rather than aligning them, and indeed improved over raw DOFA. However, its features transferred less effectively than BotaCLIP (BWiAuSclR), confirming the advantage of contrastive alignment for generalization across tasks. Taken together, these results indicate that even lightweight injections of ecological knowledge, through vegetation composition data, ecological pretext tasks (e.g., botania) and regularization, can steer generic EO embeddings toward ecologically meaningful spaces. Similar to recent multimodal ecological foundation models (Zermatten et al., 2025; Trantas et al., 2025).

The downstream evaluation provides a more direct assessment of ecological utility. Plant prediction gains (+14.9% TSS) confirm that the aligned image embeddings now encode botanical information from relevés. One might argue that this task is close to the training signal, since both rely on the same vegetation plots. However, the contrastive objective never involved binary presence/absence labels, only continuous abundance values from relevés. This makes the plant prediction task a genuine transfer: it tests whether the information injected through alignment can be easily retrieved from images alone and using simple models. In this sense, the plant task can be viewed as a sanity check rather than circularity. Butterflies (+10.4% BI), by contrast, relies on an independent dataset

and is ecologically well grounded. Given their pollinator role, diurnal butterflies' distributions are tightly linked to host plants and vegetation composition, making this task the clearest demonstration that BotaCLIP embeddings capture transferable ecological interactions. For soil trophic groups, improvements are smaller ($+1.8\%$ $\rho$), in line with reports that aboveground imagery provides weak constraints on belowground biodiversity (Cerna et al., 2025). Yet the fact that improvements exist suggests that vegetation information in images, injected by the contrastive alignement, correlates with certain soil trophic groups (e.g., bacteria, fungi, protist), providing complementary but indirect information.

Embedding visualizations and cluster metrics suggest that BotaCLIP preserves the global geometry of DOFA while enhancing its ecological semantics. UMAP plots reveal sharper boundaries among broad landscape categories, with improved Davies–Bouldin and Calinski–Harabasz scores confirming more structured clusters. These observations resonate with theoretical perspectives on contrastive learning, where the balance between alignment of positive pairs and uniformity of the embedding distribution yields representations that are both compact and diverse (Wang & Isola, 2020). By emphasizing local similarities, BotaCLIP refines fine-grained ecological distinctions without collapsing the global space.

Our approach connects to recent work on regularization for representation learning. The Three Towers model (Kossen et al., 2023) contrasts each modality with a pretrained encoder. Our regularization directly preserves similarity relations, without the need to keep high similarity with the DOFA embeddings. Ex-MCR (Zhang et al., 2024) also regularizes projected spaces to match the original one, but our extension is simpler, adding only one modality (species relevés) and through direct regularization instead of stacking contrastive terms in the loss. While finalizing this draft, we became aware of DinoV3 (Siméoni et al., 2025), which introduces a Gram anchoring loss closely related to ours. However, our formulation reweights pairs to emphasize local structure.

Beyond accuracy, BotaCLIP offers a low-cost pipeline for ecological specialization of EO foundation models. Instead of retraining large models, we adapt lightweight tabular encoders with a simple regularization term while keeping the DOFA backbone fixed. This modular strategy balances efficiency and transferability, enabling scalable biodiversity applications without prohibitive costs.

Finally, from a conceptual standpoint, contrastive alignment can be viewed as a modern extension of Canonical Correspondance Analysis (CCA) (ter Braak, 1986), long used in ecology to relate species composition to environmental gradients. While CCA projects species and environment matrices onto linear canonical axes, contrastive learning generalizes this idea by mapping heterogeneous modalities into a shared nonlinear space. This view resonates with developments in machine learning such as Deep CCA (Andrew et al., 2013; Sun et al., 2020), which similarly reinterprets CCA in a nonlinear setting. Both frameworks aim to uncover latent ecological structure, but contrastive alignment better scales to high-dimensional imagery, exploits weak supervision from paired data, and transfers well to new tasks. In this sense, BotaCLIP can be interpreted as a nonlinear, multimodal analogue of CCA, where vegetation relevés anchor remote-sensing embeddings into ecologically meaningful dimensions.

# 6 CONCLUSION

We introduced BotaCLIP, a lightweight framework that adapts the EO foundation model DOFA by aligning aerial orthophotos with in-situ vegetation relevés through contrastive learning. Our regularization mitigates catastrophic forgetting, preserving DOFA's broad representations while injecting ecological semantics. Across three downstream tasks (plants, butterflies, and soil trophic groups), BotaCLIP consistently outperformed raw DOFA and supervised baselines, demonstrating lightweight domain adaptation as an effective alternative to costly end-to-end retraining.

Beyond biodiversity, this approach illustrates how domain-specific knowledge can adapt foundation models in data-scarce sciences. Future work includes extending BotaCLIP to other ecological modalities (traits, acoustics) and exploring tri-modal alignments of images, relevés, and environmental covariates, with potential applications in agriculture and forestry to build frugal, ecologically informed embeddings.

REPRODUCIBILITY STATEMENT

All details about the architecture, loss function, and training strategy are provided in Section 3, with further information in the appendix. Code to reproduce the experiments will be released upon publication. The vegetation, butterfly, and soil datasets are derived from existing ecological surveys and will be shared in processed form subject to licensing constraints.

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

# A APPENDIX

## A.1 DETAILS OF ABLATION STUDY SETUPS

For reproducibility, we provide details of the ablation experiments summarized in Table 1 and statistically analyzed in Table 2. Each variant is identified by a compact code that concatenates three components:

- **Architecture:** B = Botania encoder, M = MLP adapter, A = Attention-based adapter.
- **Augmentation:** WiAu = trained with image augmentation, WoAu = trained without augmentation.
- **Objective:** Scl = sigmoid contrastive loss, SclR = regularized sigmoid contrastive loss.

For example, *BWiAuSclR* denotes the Botania encoder with augmentation and the regularized loss, while *MWoAuScl* refers to an MLP adapter without augmentation under the plain loss.

**Architectural variants.** We explored three encoder designs for the tabular branch of BotaCLIP:

- **MLP (MWiAuScl, MWoAuScl):** a lightweight two-layer multilayer perceptron. Tabular inputs are passed through a linear layer ($d_{tab} \rightarrow 1024$), ReLU activation, dropout (0.1), and a second linear layer projecting to the shared embedding space ($1024 \rightarrow 768$). The image branch follows a similar structure, mapping DOFA embeddings through a linear layer ($d_{img} \rightarrow 2600$), ReLU, dropout (0.1), and projection to 768 dimensions. Both image and tabular projections are $\ell_2$-normalized before computing contrastive loss.

- **Attention (AWiAuScl, AWoAuScl):** extends the MLP variant by inserting a 4-head Multihead Attention block on the tabular branch. The species–cover vector is first reduced linearly ($d_{tab} \rightarrow 1024$), followed by LayerNorm, Multihead Attention, residual connection, and a second LayerNorm. The attended features are then passed through ReLU, dropout (0.1), and a linear projection ($1024 \rightarrow 768$). This design aims to capture interactions among heterogeneous ecological features beyond simple feed-forward transformations. The image branch is identical to the MLP variant.

- **Botania (BWiAuSclR, BWoAuSclR):** the pre-trained Botania encoder as tabular branch, combined with a linear adapter as described in 2.1.

**Data augmentation.** WiAu variants apply image-side augmentations (random flips, rotations, resized crops, brightness/contrast jitter, Gaussian blur/noise) before feature extraction with DOFA. WoAu variants use raw image tiles without augmentation.

**Loss functions.**

- **Scl:** original sigmoid contrastive loss.

- **SclR:** proposed regularized version, preserving local similarity relations of DOFA embeddings.

**Training details.** All models were trained with AdamW ($lr = 10^{-3}$, weight decay $= 10^{-3}$), batch size 256, and early stopping with patience 10. The DOFA image encoder was frozen, avoiding recomputation of patch-level embeddings. Lightweight adapters and the tabular branch were updated in all variants. For Botania-based models, the tabular encoder was initialized from its pre-trained checkpoint but remained trainable, allowing it to adapt jointly with the contrastive objective. Projection dimension was fixed to 768.

**Full experiment list.** In total, we evaluated six configurations: *MWiAuScl*, *MWoAuScl*, *AWiAuScl*, *AWoAuScl*, *BWiAuSclR*, *BWoAuSclR*. These correspond to the most representative axes of variation—architecture, augmentation, and loss design, and are the ones reported in the main text. In practice, we explored a broader set of runs, including additional architectural choices, batch sizes, regularization strategies, random seeds, spatial partitions, and ratios of pseudo-absences (e.g. in butterflies). We restrict reporting to these six canonical settings to provide a concise yet comprehensive picture of how each design axis influences performance.

**Statistical analysis.** Table 2 reports the outcome of statistical tests across ablation experiments. For each dataset, we first ran a Friedman test to verify whether performance differences across models were significant. We then compared the best configuration (BWiAuSclR) against all alternatives using paired Wilcoxon signed-rank tests at the per-species (plants, butterflies) or per-group (soil) level, applying Holm–Bonferroni correction. Reported values include the Wilcoxon statistic, adjusted $p$-values, median differences, and relative changes. In all cases, the Friedman test detected significant global differences. Pairwise tests confirm that BWiAuSclR outperforms DOFA on plants, butterflies, and soil, with larger effect sizes for plants and butterflies. Comparisons against other BotaCLIP variants highlight that augmentation and the Botania encoder are the main drivers of improvement.

Table 2: Statistical analysis of ablation experiments. For each dataset we report the best model, Friedman test results, and Wilcoxon-Holm pairwise tests (Wilcoxon statistic, $p$-value, median difference, and relative change).

| Dataset | Comparison | Wilcoxon stat | $p$-value | Median diff | % change |
|---|---|---|---|---|---|
| Plant (TSS) | BWiAuSclR vs DOFA | 0.0 | $2.8 \times 10^{-20}$ | 0.0649 | +14.9% |
| | BWiAuSclR vs BWoAuSclR | 2533 | $4.9 \times 10^{-2}$ | 0.0031 | +0.6% |
| | BWiAuSclR vs MWiAuScl | 1.0 | $2.9 \times 10^{-20}$ | 0.0712 | +16.6% |
| | BWiAuSclR vs MWoAuScl | 15.0 | $4.2 \times 10^{-20}$ | 0.0510 | +11.4% |
| | BWiAuSclR vs AWiAuScl | 0.0 | $2.8 \times 10^{-20}$ | 0.0896 | +21.9% |
| | BWiAuSclR vs AWoAuScl | 0.0 | $2.8 \times 10^{-20}$ | 0.0763 | +18.0% |
| | Friedman stat = 501.5, $p = 3.9 \times 10^{-105}$ 
 Best = BWiAuSclR | | | | |
| Soil (Spearman $\rho$) | BWiAuSclR vs DOFA | 241.0 | $7.6 \times 10^{-5}$ | 0.0088 | +1.8% |
| | BWiAuSclR vs BWoAuSclR | 655.5 | 0.944 | 0.0049 | +1.0% |
| | BWiAuSclR vs MWiAuScl | 550.5 | 0.401 | 0.0139 | +2.9% |
| | BWiAuSclR vs MWoAuScl | 601.0 | 0.561 | 0.0084 | +1.7% |
| | BWiAuSclR vs AWiAuScl | 635.5 | 0.797 | 0.0171 | +3.6% |
| | BWiAuSclR vs AWoAuScl | 245.0 | $8.9 \times 10^{-5}$ | 0.0124 | +2.6% |
| | Friedman stat = 28.0, $p = 9.3 \times 10^{-5}$ 
 Best = BWiAuSclR | | | | |
| Butterfly (BI) | BWiAuSclR vs DOFA | 918.0 | $1.2 \times 10^{-15}$ | 0.0688 | +10.4% |
| | BWiAuSclR vs BWoAuSclR | 4332 | 0.672 | 0.0144 | +2.0% |
| | BWiAuSclR vs MWiAuScl | 89.0 | $7.1 \times 10^{-23}$ | 0.1269 | +21.1% |
| | BWiAuSclR vs MWoAuScl | 130.0 | $1.8 \times 10^{-22}$ | 0.0954 | +15.1% |
| | BWiAuSclR vs AWiAuScl | 23.0 | $1.6 \times 10^{-23}$ | 0.1670 | +29.7% |
| | BWiAuSclR vs AWoAuScl | 269.0 | $3.5 \times 10^{-21}$ | 0.0943 | +14.9% |
| | Friedman stat = 509.2, $p = 8.8 \times 10^{-107}$ 
 Best = BWiAuSclR | | | | |

