# OpenReview forum: "BotaCLIP: Contrastive Learning for Botany-Aware Representation of Earth Observation Data"
_ICLR.cc/2026/Conference — ICLR 2026 Conference Withdrawn Submission_

### Official Review · Reviewer_QPib · 2025-10-26

**Soundness:** 2
**Presentation:** 2
**Contribution:** 1
**Rating:** 0
**Confidence:** 4

**Summary:**

The paper introduces BotaCLIP which repurposes DOFA to be botany-aware. The authors use frozen DOFA encoder along with a learnable adapter on top and align its embeddings with in-situ vegetation survey data. The authors add a regularization loss to prevent forgetting and maintain the manifold structure of DOFA's embedding space. The authors conduct experiments on plant species abundance prediction, butterfly prediction and soil trophic group prediction.

**Strengths:**

1. The paper is easy to understand and proposes a simple contrastive-based technique to adapt DOFA to be botany-aware with a regularization loss to maintain the original DOFA embeddings space.

**Weaknesses:**

1. I believe the paper has very limited technical novelty given there are several works already aligning satellite image representations with wildlife observations and descriptions. They authors should compare their work with WildSat, TaxaBind and EcoWikiRS and clearly discuss how their work is technically different from these works.
2. In terms of the method itself, the paper uses existing architectures and losses. The authors could have explored some domain-specific design choices such as handling spatio-temporal biases in the data. The authors could have added geolocation or other metadata into their framework which is commonly done by other works.
3. How about LoRA fine-tuning of DOFA for this task. How would that perform as compared to the proposed regularization based approach?
4. Why contrastive-based training? Why not model the task as a supervised learning problem that predicts the species abundance given a satellite image? Then one can extract the satellite embeddings from the penultimate layer.
5. Why DOFA? Very limited experiments on the choice of pretrained-encoder model. The paper only uses DOFA as the base model but could have explored other multimodal encoders such as AnySat or Galileo.
6. No baseline results reported using existing general-purpose satellite image encoders. The authors should atleast compare with models like Galileo, AnySat or SatMAE. I also encourage the authors to compare with more task specific models such as WildSat, TaxaBind and EcoWikiRS.
7. Since model was trained using vegetation surveys, why does it perform well in butterfly prediction? Also, how does the model generalize to different regions of the world?

Suggestions:
The paper falls short of several key experiments and analysis and lacks explanation on several design choices. The paper does not report any baseline other than the models trained using DOFA. This raises concerns whether the proposed model is truly state-of-the-art for the task.

**Questions:**

Please see weaknesses.

---

### Official Review · Reviewer_5djk · 2025-10-29

**Soundness:** 2
**Presentation:** 2
**Contribution:** 2
**Rating:** 2
**Confidence:** 4

**Summary:**

The authors propose BotaCLIP, a framework that adapts an Earth Observation foundation model called DOFA using aerial imagery and botanical data. The model is trained in a contrastive manner using vegetation survey data and another MLP model called Botania. The loss function is based on two compenents, a stadard contrastive loss (2) and a regularization term (3) that enforces the pairwise similarities of the embeddings to be close to the ones of the DOFA ouput. The authors also show how to make use of this model for three downstream tasks (plant monitoring, insect monitoring, soil monitoring), given a data-scarce learning setting. The overall model is based on DOFA and Botania, along with two simple linear embedding layers. The authors provide details related to the training strategy (e.g., initialization of one of layers as identity) and an experimental evaluation related to the three downstream tasks.

**Strengths:**

- Interesting application domain
- A model architecture suited for the task
- The paper is generally well written; I could follow most of the parts easily (except for the ablation study)

**Weaknesses:**

- Lack of novelty: My main concern is a lack of novelty. The paper depicts a decent work, but is based on well-known techniques. The application is an interesting one, but overall, the contribution is not sufficient to meet the very high standards of ICLR in my opinion.
- Results: The experimental comparison is only based on DOFA embeddings and BotaSP, and the improvement over these baselines is relatively small (e.g., for Plant, the BWiAuSclR-F1 scores are close to the simple BotaSP baseline)

**Questions:**

I could follow the description of your approach easily, so I do not have any technical questions.

Suggestion: I find the description of the different variants a bit confusing. I would suggest to only focus on the best-performing model(s) in the end (e.g., on BWiAuSclR and BWoAuSclR). They are not really inferior compared to the other baselines, which could be moved to the appendix.

---

### Official Review · Reviewer_rwpp · 2025-10-31

**Soundness:** 3
**Presentation:** 3
**Contribution:** 2
**Rating:** 4
**Confidence:** 4

**Summary:**

This paper introduces BotaCLIP, a multimodal contrastive framework that adapts the pre-trained DOFA Earth Observation foundation model by aligning aerial imagery with botanical releves. The key innovation is a regularization strategy that preserves DOFA's local similarity structure while injecting ecological knowledge through contrastive learning.

**Strengths:**

1. Adapting foundation models to specialized domains (e.g., in data-scarce fields like Ecology) without expensive retraining is highly relevant. The paper clearly articulates why generic EO embeddings are insufficient for fine-grained biodiversity modeling.
2. This paper evaluates across three ecologically diverse tasks spanning different taxonomic groups (plants, insects, soil) and prediction types (classification, habitat suitability, abundance regression), which demonstrates meaningful transferability.
3. Systematic exploration of architectural variants (Botania/MLP/Attention), augmentation strategies and loss functions effectively isolates the contribution of each component.
4. The link to Canonical Correspondence Analysis and careful treatment of ecological concepts (pseudo-absences, Boyce Index, spatial buffers) demonstrates domain expertise.

**Weaknesses:**

1. The approach essentially applies CLIP-style contrastive learning with a similarity-preserving regularization term. It represents an incremental contribution rather than a methodological improvement. Additionally, the authors acknowledge that DinoV3 introduces a closely related "Gram anchoring loss,'' which raises questions about the novelty of the contribution.
2. The $+1.8$\% improvement in Spearman's $\rho$ for soil monitoring (Table-1) might look significant but practically small. While the authors acknowledge that aboveground imagery weakly constrains belowground biodiversity, this limits the impact for certain applications. More discussion of when BotaCLIP provides substantial vs. marginal gains would strengthen the paper.
3. The result section only compares against DOFA and BotaSP. There is no comparison with other parameter-efficient adaptation methods (LoRA, adapters, prompt tuning), multimodal EO foundation models, TIP (Du et al., 2024) which also aligns tabular and image data.
4. Despite claiming to be "lightweight'' and "inexpensive,'' the paper provides no runtime, memory usage, or FLOPS comparisons. How much faster/cheaper is BotaCLIP than alternative methods? This is essential for evaluating practical deployment.
5. The regularization weight $\lambda = 1$ is fixed throughout all experiments without ablation studies or justification. Similarly, the identity initialization for the image adapter is presented as a design decision, but no sensitivity analysis demonstrates whether this initialization strategy is valid.

**Questions:**

1. How sensitive are the results to the regularization weight $\lambda$ or temperature $\tau$? An ablation over $\lambda \in$ {0.1, 0.5, 1, 2, 5, 10} would strengthen the paper.
2. How does performance scale with the amount of releve data? What is the minimum number of samples needed for effective adaptation?
3. Can the authors provide runtime, memory usage, and FLOPS comparisons with (a) supervised pretraining (BotaSP), (b) end-to-end DOFA fine-tuning and (c) no adaptation?
4. Have you evaluated data from regions outside the French Alps? Even a small-scale demonstration would address concerns about generalization. Additionally, using GradCAM or attention maps to show which image regions drive predictions for different species, will provide ecological interpretability.
5. Why not compare with other parameter-efficient adaptation methods (LoRA, prompt tuning, PEFT) that are standard in the foundation model literature?
6. Regarding Section 4.2 and Figure 3, the UMAP 2D visualizations appear difficult to interpret. Both subplots show heavily overlapping points with similar spatial boundaries. Could you clarify what specific visual patterns we should observe in Figure 3 that demonstrate BotaCLIP's advantages? Would alternative visualizations (e.g., per-category cluster tightness, confusion matrices between categories) more effectively communicate the improvements?

---

### Note · Authors · 2025-11-20

**Comment:**

Dear Area Chair and Reviewers,


We would like to thank you for your time and constructive feedback on our submission.
After carefully considering the reviews and the decision, we have decided not to resubmit the manuscript for the rebuttal or revision phase.


We sincerely appreciate the reviewers’ comments and suggestions, which will help us improve the work for a future submission to another venue.


Thank you again for your efforts and for the opportunity to present our work to the ICLR community.


Kind regards,

Selene Cerna

on behalf of the authors

**Withdrawal Confirmation:**

I have read and agree with the venue's withdrawal policy on behalf of myself and my co-authors.